# Decision Tree Analyses to Explore the Relevance of Multiple Sex/Gender Dimensions for the Exposure to Green Spaces: Results from the KORA INGER Study

**DOI:** 10.3390/ijerph19127476

**Published:** 2022-06-18

**Authors:** Lisa Dandolo, Christina Hartig, Klaus Telkmann, Sophie Horstmann, Lars Schwettmann, Peter Selsam, Alexandra Schneider, Gabriele Bolte

**Affiliations:** 1Department of Social Epidemiology, Institute of Public Health and Nursing Research, University of Bremen, 28359 Bremen, Germany; hartig@uni-bremen.de (C.H.); telkmann@uni-bremen.de (K.T.); sophie.horstmann@uni-bremen.de (S.H.); gabriele.bolte@uni-bremen.de (G.B.); 2Health Sciences Bremen, University of Bremen, 28359 Bremen, Germany; 3Institute of Health Economics and Health Care Management, Helmholtz Zentrum München, German Research Center for Environmental Health (GmbH), 85764 Neuherberg, Germany; lars.schwettmann@helmholtz-muenchen.de; 4Department of Economics, Martin Luther University Halle-Wittenberg, 06108 Halle (Saale), Germany; 5Department Monitoring and Exploration Technologies, Helmholtz Centre for Environmental Research GmbH—UFZ, 04318 Leipzig, Germany; peter.selsam@ufz.de; 6Institute of Epidemiology, Helmholtz Zentrum München, German Research Center for Environmental Health (GmbH), 85764 Neuherberg, Germany; alexandra.schneider@helmholtz-muenchen.de

**Keywords:** sex, gender, intersectionality, recursive partitioning, subgroup analysis, greenness, normalized difference vegetation index (NDVI)

## Abstract

Recently, attention has been drawn to the need to integrate sex/gender more comprehensively into environmental health research. Considering theoretical approaches, we define sex/gender as a multidimensional concept based on intersectionality. However, operationalizing sex/gender through multiple covariates requires the usage of statistical methods that are suitable for handling such complex data. We therefore applied two different decision tree approaches: classification and regression trees (CART) and conditional inference trees (CIT). We explored the relevance of multiple sex/gender covariates for the exposure to green spaces, measured both subjectively and objectively. Data from 3742 participants from the Cooperative Health Research in the Region of Augsburg (KORA) study were analyzed within the INGER (Integrating gender into environmental health research) project. We observed that the participants’ financial situation and discrimination experience was relevant for their access to high quality public green spaces, while the urban/rural context was most relevant for the general greenness in the residential environment. None of the covariates operationalizing the individual sex/gender self-concept were relevant for differences in exposure to green spaces. Results were largely consistent for both CART and CIT. Most importantly we showed that decision tree analyses are useful for exploring the relevance of multiple sex/gender dimensions and their interactions for environmental exposures. Further investigations in larger urban areas with less access to public green spaces and with a study population more heterogeneous with respect to age and social disparities may add more information about the relevance of multiple sex/gender dimensions for the exposure to green spaces.

## 1. Introduction

Sex/gender with its multiple biological and social dimensions has not yet been adequately considered in environmental health research [1,2,3,4]. To comprehensively assess the impact of gender relations as well as sex-linked biology, there has been a call to integrate sex/gender with its complexity and entanglement of biological and social dimensions into health research beyond a simple binary approach [5,6,7,8]. We use the term “sex/gender” to express this entanglement of sex and gender [9,10]. Moreover, an intersectionality perspective strengthens the consideration of structural causes of health inequities such as systems of power and discrimination processes [11,12].

To meet the challenge of a comprehensive integration of sex/gender into environmental health research, various data to depict multidimensionality, variety, embodiment and intersectionality in the study of sex/gender impacts [9] have to be used in multivariate statistical analyses. In quantitative health research, standard methods to examine relationships between covariates and outcomes are regression models. These are, however, limited to a small number of covariates and are not flexible enough to uncover unspecified, complex, and non-linear covariate-outcome relationships [13,14]. A family of methods to analyze complex and high-dimensional data are decision trees, also called recursive partitioning. Decision trees are exploratory, non-parametric methods that recursively partition a sample into subgroups based on covariate values, thereby classifying individuals into subgroups that are homogeneous with respect to the main outcome [14,15]. Decision trees have also been identified as particularly useful methods for research based on intersectionality theory [16,17,18] as they can identify complex and unsuspected interactions between covariates, even if they are non-linear [16].

Although decision trees are not as widely used as regression models, researchers have been encouraged to use them in epidemiological or public health research [14,15,19]. In recent research, decision trees have therefore been used to identify subgroups with homogeneous health outcomes [17,20,21,22] or health-related behaviors [13,16,18]. Mena [17] for instance, used an intersectional-informed approach to find subgroups that show a high prevalence of frequent mental distress. While Winkler [18] applied decision trees to investigate the involvement in physical activity and sleep in a young adult population.

As an exemplary thematic field for the comprehensive integration of sex/gender into environmental health, we chose exposure to green spaces in the residential environment. Within environmental health research, green spaces and green infrastructure have been proven to be an important environmental resource of health [23,24,25,26]. Green spaces act as places for socialising, exercise and recreation [23,27,28] and can have a positive impact on physical activity as well as social and psychological well-being [29]. Green spaces also reduce exposure to noise, air pollutants and intense heat and improve air quality [25,27,29].

In recent years evidence increased that social inequalities exist in availability of, or access to, green spaces. Evidence suggests that population groups with lower socioeconomic positions have less access to environmental resources like green spaces [30,31,32]. However, sex/gender—as an important social determinant of health—is not often analyzed in studies on social inequalities in exposure to green spaces [31]. A few studies discussed differences in the usage of public green spaces and parks by a binary sex/gender category [33,34,35], without considering sex/gender differences in the exposure to green spaces.

The aim of our study was twofold: firstly, we wanted to test whether decision trees are suitable methods to analyze complex data when integrating multiple sex/gender dimensions into environmental research. Currently there are debates about advantages and disadvantages of different types of decision trees [15,36]. Referring to these, we compared two types of decision trees: the most often used classification and regression trees (CART) [37,38] and a prominent alternative, conditional inference trees (CIT) [39,40]. Secondly, by applying decision tree methods, we aimed to identify and describe homogeneous subgroups with respect to exposure to green spaces, considering a large range of sex/gender covariates simultaneously and their possible interactions.

## 2. Materials and Methods

### 2.1. INGER Project

The focus of the collaborative research project INGER (https://www.uni-bremen.de/en/inger, accessed on 25 May 2022) was to integrate sex/gender into environmental health research. The project aimed to develop innovative methods for data collection and analyses in population-based studies on environmental health, to be able to assess the impacts of sex/gender more comprehensively. Within INGER, we developed a multidimensional sex/gender concept with a focus on intersectionality [9] and created new sex/gender questionnaire modules to improve sex/gender data collection in population-based studies. We used this newly developed INGER questionnaire within the Cooperative Health Research in the Region of Augsburg (KORA) study to investigate the aforementioned research questions.

### 2.2. Study Population

The research platform KORA was designed to evaluate the links between health, disease and the living conditions of the population of Augsburg and two adjacent counties [41]. Since 1984, four cross-sectional surveys at intervals of 5 years and various follow-ups have been conducted. The four surveys were S1 (1984/85, participants born between 1920 and 1959), S2 (1989/90, 1915–1964), S3 (1994/95, 1920–1969) and S4 (1999–2000, 1925–1974). In 2019, the paper-based INGER questionnaire was sent to 5256 eligible KORA participants aged 44–93 years. These participants included all participants of the KORA FIT study, which was conducted in 2018/2019 and to which participants of all four surveys with a current age of 54–75 years were invited. In addition, the INGER questionnaire was sent to all younger participants of S3 (49–53 years) and to all other participants of S4 who had not participated in KORA FIT (44–53 and 74–93 years). Within the INGER survey, participants answered the newly developed sex/gender questionnaire module as well as an extensive set of questions about residential green spaces.

### 2.3. Sex/Gender Covariates

For our main decision tree analyses we chose a total of 40 sex/gender covariates based on our INGER multidimensional sex/gender concept [9]. This concept describes an individual sex/gender self-concept that is embedded in an environment and society that is defined by structural sex/gender relations. When operationalizing this concept, 17 covariates represent the individual sex/gender self-concept: one covariate each for the dimensions *sex assigned at birth* and *current sex/gender identity*, twelve covariates operationalizing the dimension *internalized sex/gender roles*, and three covariates for the dimension *externalized sex/gender expressions*. We included 23 covariates that contribute to explain the structural sex/gender relations: nine covariates corresponding to the *experience of discrimination*, eight covariates related to *care activities*, and six covariates corresponding to *intersectionality-related social categories*. For a complete list of the 40 covariates, including the questions asked and possible answer categories for each question see Table 1.

As some variables included in the KORA FIT survey can add some additional information that can be interpreted with respect to our INGER multidimensional sex/gender concept [9], we decided to run additional decision tree analyses for all INGER participants who had also taken part in the KORA FIT study with a total of 53 covariates, i.e., the original 40 covariates plus 13 additional ones. Of the 13 new covariates, six correspond to *intersectionality-related social categories*, three portray *health related behaviors* and four covariates are *psychosocial factors*. For a complete list of the 13 extra covariates, including the questions asked within the KORA FIT survey, as well as the possible answer categories for each question see Appendix A.

### 2.4. Exposure Variables

We operationalized green spaces in two different ways. Firstly, we measured the subjectively reported access to public green spaces and the quality of these public green spaces. Secondly, we considered the general greenness in the residential environment, measured both subjectively and objectively. The subjective measurements were based on our KORA INGER survey and the objective measurements comprise *Normalized Difference Vegetation Index* (NDVI) data available for the participants’ residential address. Overall, we ran decision tree analyses for five different exposure variables.

#### 2.4.1. Access to Public Green Spaces (Subjectively Measured)

Participants were asked whether they had access to public green spaces in their residential environment, naming parks, forests, or meadows as examples (one variable).

#### 2.4.2. Access to High Quality Public Green Spaces (Subjectively Measured)

This exposure measure with three answer categories (*access to high quality public greenspaces, access only to lower quality public greenspaces, no access to public green spaces*), was based on a combination of three questions: the first question was the one described above about the access to public green spaces. The participants who answered no to this question were classified as having *no access to public green spaces*. The participants who answered yes to this question were further divided into having *access to high quality public green spaces* vs. *access only to lower quality public green spaces* based on their response to two follow-up questions about the quality of available green spaces. In these follow-up questions, participants were asked to indicate on a five-point scale from strongly agree to strongly disagree, whether the green spaces in their residential environment were well maintained, and secondly whether the green spaces in their residential environment were of high quality. We classified participants within the category *access to high quality public green spaces* when they gave the answer “strongly agree” to both above statements, and all other remaining participants were categorized as having *access only to lower quality public green spaces*.

#### 2.4.3. Greenness in the Residential Environment (Subjectively Measured)

Participants were asked how green their neighborhood was, considering every type of greenspace, from grass verges on the streets to gardens and parks. The original question had four answer categories (*very green, a little green, hardly green, not green at all*). The last two categories had a very small sample size (*hardly green* = 69, *not green at all* = 9), so these two were grouped together as *hardly green*.

#### 2.4.4. Greenness within a 300 or 1000 m Buffer around the Residential Address (Objectively Measured)

For these two exposure variables we used NDVI data, calculated for the year 2019. NDVI is a measure of vegetation density, i.e., greenness [42]. We used Landsat 8 Operational Land Imager (OLI) satellite images with a resolution of 30 m and Sentinel-2 images with 10 m ground resolution with less than 1% cloud cover for single images. Image and quality selection as well as all calculations were performed using Google’s Earth Engine Code Editor (https://code.earthengine.google.com, accessed on 4 October 2021). Images were collected during April to October and calculated as median of all accepted images relying on the atmospherically corrected reflectance of near-infrared (NIR) and visible red (RED) light, using the standard formula: NDVI  =  (NIR − RED)/(NIR  +  RED). NDVI values comprise a possible value range from −1 to +1 with values about 0 and lower indicating rock, sand, snow, water and densely urbanized areas, values near +1 a very high density of photosynthetic active plants [43,44]. Negative values were set to NAs (missing values). Water areas were masked according to the Copernicus Global Land Cover Map [45]. For each participant, the NDVI values within a certain buffer around their residential address were averaged. We chose two different buffers, a 300 m, and a 1000 m buffer, the first representing the area one can easily access by foot and the second a more extended area as a direct comparison.

### 2.5. Recursive Partitioning

Recursive partitioning is a statistical method used for subgroup analysis. A decision tree is grown, which provides a partitioning of the sample population into several subgroups based on dichotomized independent variables. The root node contains the entire sample and is depicted at the top of a tree. A split produces two mutually exclusive subnodes determined by a splitting rule. Each split is induced on a single independent variable and aims to maximize the homogeneity of the target variable within the subgroups. This procedure is carried out recursively at each subnode until some stopping criterion is met. Nodes without successors are called leaves or terminal nodes. Edges resemble decision rules. For more details and applications, we refer to [14,15,19] and references therein. Different algorithms employ different stopping criteria and metrics to measure homogeneity. In this article we focus on two types of recursive partitioning algorithms, namely CART [37] and CIT [39]. A major advantage of both of these decision tree methods compared to e.g., standard regression models is that they have an intrinsic mechanism to handle missing values in the covariates. A complete case analysis is not necessary. In a first step, where the best splitting variable is chosen, observations that have missing values in the currently evaluated covariate are ignored. Observations with missing values are then assigned to child nodes by so-called surrogate splits, which mimic the decision rule as closely as possible [46].

### 2.6. Classification and Regression Trees (CART)

CART, developed by Breiman [37], refers to two types of decision tree algorithms: classification trees are applied when the target variable is categorical, whereas regression trees cover numerical outcomes.

#### 2.6.1. Splitting Criteria

CART is a greedy algorithm and searches for the best split among all permissible splits. Classification trees measure the homogeneity in a node (or the lack thereof) by misclassification error, information gain or Gini impurity. In contrast, regression trees minimize the variance. In both cases the quality of a split is then determined by averaging these measures for both resulting subnodes. In our analyses, classification trees using Gini impurity are grown to analyze the three subjective exposure measures while regression trees are applied to the NDVI exposure variables.

#### 2.6.2. Stopping Criteria

The algorithm terminates if one of several possible stopping criteria is met. A common rule is to specify a minimum number $n_{min}$ of observations in a node. A split is only induced if both resulting child nodes contain at least $n_{min}$ observations. This is often set to about 1% of the sample size [19]. However, it has been recommended to use no less than 50 observations as a minimum bound, as terminal nodes with less observations lack statistical robustness and have little predictive power [47]. Therefore, we set the minimum number of observations allowed in a node to 50. This is also comparable to a recent study that had a similar sample size and used the same threshold [22]. Another often used criterion is to bound the depth of the tree, i.e., the maximum number of consecutive splits. In our analyses we restricted the maximal depth of the tree to four.

#### 2.6.3. Pruning

Despite the above-mentioned stopping criteria, tree-based methods are still prone to overfitting. This means that the tree becomes too large such that the derived decision rules are too specific and may not be applicable to other data [14]. Therefore, it is common to prune the tree until only meaningful subgroups are left. Breiman [37] suggests applying cross-validation to select the best subtree. They recommend identifying the tree with the smallest cross-validation error and then adding its corresponding standard error (SE). The smallest tree within this range is said to follow the so-called *1-SE rule* and is deemed the best-sized subtree. However, it has recently been shown that this *1-SE rule* is rather conservative with quite a low type one error [15]. An alternative, less conservative pruning rule is to choose the subtree corresponding to the minimum cross-validation error [15,16,36,48]. Here, we will present the CART trees based on this less conservative pruning rule, but will also describe whether the more conservative pruning based on the *1-SE rule* would have led to the same tree. In *rpart* one can also set an a-priori complexity parameter (cp) value to avoid CART to grow overly large trees that are then always pruned back in the next step. The default value in *rpart* is cp = 0.01, but this sometimes leads to over-pruning in larger datasets missing out on meaningful splits [38]. We therefore set the cp value to a less conservative 0.001 and performed pruning as described above.

#### 2.6.4. Variable Importance

CART also calculates a variable importance measure that indicates how important the covariates were in the splitting process. This is performed by considering all occurrences at which the covariate appears, either as a primary or as a surrogate splitting variable. This measure can help to identify variables which are not represented in splits because they are masked by other variables, possibly due to collinearity. The variable importance values across all covariates are scaled to sum up to 100 [38].

### 2.7. Conditional Inference Trees (CIT)

CIT is a decision tree method developed by Hothorn [39]. The major difference to CART is that feature selection and the actual splitting process are separated. Moreover, it utilizes a concept of statistical significance to determine splitting variables.

#### 2.7.1. Splitting Criteria

To avoid an exhaustive search, a *p*-value driven feature selection step searches for the best splitting variable. More specifically, in the first step, for each covariate the null-hypothesis of independence of the target variable is tested at some prespecified significance level α. The covariate with the strongest bivariate association with the dependent variable is selected. In a second step, the algorithm identifies the optimal binary partition based on the selected covariate. This two-step approach enables an unbiased selection procedure overcoming a problem that is often encountered in CART: the bias to select variables with many possible splits or missing values [39].

#### 2.7.2. Stopping Criteria

In addition to the stopping criteria of a minimum node size of 50 and a maximum depth of four described above, the algorithm also terminates if none of the covariates show a significant association with the target variable. In our analysis we set the *p*-value threshold to 0.05 and used a Bonferroni correction to adjust for multiple testing. Moreover, for CIT it is not considered necessary to prune the trees in a following step [20].

#### 2.7.3. Random Forests to Calculate Variable Importance

One problem with using decision trees is that they can be vulnerable to random patterns in the data, and results may thus not always be reproducible in slightly different samples. It is therefore advisable to calculate the variable importance measure using random forests in order to evaluate whether the splitting variables found in the original CIT also have the highest variable importance values across an ensemble of trees [18,46]. Therefore, a random forest consisting of 500 trees was grown. Following a rule of thumb, we used six randomly selected covariates (i.e., the square-root of the overall number of covariates) for classification and 13 (i.e., one third of the overall number of covariates) in regression for each tree within the forest [49]. Variable importance was measured in terms of mean decrease in classification accuracy. Therefore, a covariate is randomly permuted over all trees breaking up the association with the response variable. If this covariate is associated with the response, prediction accuracy should decrease when using the permuted covariate to predict the response (see Strobl [50] for more details). For the interpretation of the resulting variable importance measures, Strobl [46] suggested selecting those covariates as informative that have a positive value, which is higher than the random variation around zero, i.e., positive values greater than the absolute value corresponding to the lowest negative value. We refer to Strobl [46] for a more detailed explanation.

### 2.8. Software

All analyses were performed in R using version 4.0.2. CART is implemented in the *rpart* package [38]. For the CIT analysis we used the *ctree* and *cforest* functions in the *partykit* package [40].

## 3. Results

### 3.1. Sample Characteristics

From 5256 KORA participants we received 3742 valid questionnaires, corresponding to a response rate of 71.2%. Of these, 2624 participants (70.1% of the whole INGER sample) were part of the KORA FIT study. The distributions of all 40 sex/gender covariates, which were included in our analyses are presented in Table 1 (see Appendix A for distributions within the INGER KORA FIT sample). At birth, 53.9% were assigned a female sex and 45.0% a male sex, with 1.2% not answering this question. When asked about their current sex/gender identity, 53.4% of the participants classified themselves as female, 44.1% as male, 0.1% as trans, 0.1% as an identity that was not mentioned within the survey, and 0.4% did not want to be classified as any sex/gender category. As we also allowed multiple answers to this question, 0.1% of the participants answered with both female and that they did not want to classify as any sex/gender category, 0.2% answered with both male and that they did not want to classify as any sex/gender category, and 0.1% classified themselves as both male and an identity not mentioned within the survey. Furthermore, 1.6% did not give an answer to this question.

For exposure distributions within the INGER sample see Table 2 (see Appendix A for distributions within the INGER KORA FIT sample).

A further description of the INGER study population is presented in Table 3 (see Appendix A for distributions within the INGER KORA FIT sample). As only adults aged 45 or above took part in the survey, the mean age of the participants in the whole sample was 63.41 years and 47.7% were retired. Notably, the majority of participants had access to outdoor spaces, for example, 69.9% had access to a private garden, and a further 9.7% had access to a garden that they share with several parties. Furthermore, 71.7% had access to a balcony or terrace. Combining the answers to these questions showed that only 1.7% of the participants had no access to outdoor spaces at all, i.e., they had neither a garden nor balcony or terrace.

### 3.2. Access to Public Green Spaces (Subjectively Measured)

#### 3.2.1. CART Results

Running a CART analysis for the exposure variable *access to public green spaces* with the 40 sex/gender covariates did not yield any splits, even though we used a quite low cp value of 0.001 (see Appendix A, pp. 2–4).

#### 3.2.2. CIT Results

Running a CIT analysis for the exposure variable *access to public green spaces* with the 40 sex/gender covariates resulted in a tree with two splits and three subgroups based on the covariates *SGRelationsIncome* and *SGIdentity*. Importantly, however, the *cforest* analysis with 500 trees showed that none of the covariates should be further investigated according to the variable importance score threshold suggested by Strobl [46], thus the identified tree has to be interpreted with caution (see Appendix A, pp. 4–6).

### 3.3. Access to High Quality Public Green Space (Subjectively Measured)

#### 3.3.1. CART Results

Running a CART analysis for the exposure variable *access to high quality public green spaces* with the 40 sex/gender covariates yielded a tree with three splits and four subgroups (see Appendix A, pp. 7–11) after pruning with the *minimum cross-validation error rule* (pruning with the *1-SE rule* would not have led to any splits). The first split was initiated by the participant’s experiences with age discrimination (*DiscriminationAge*). Those who had already been discriminated against because of their age formed the subgroup with least *access to high quality public green spaces*. The remaining participants were split again into those with a very good self-rated financial situation and those who rated their own financial situation as bad, moderate, or good (*SGRelationsIncome*). The first group showed a higher prevalence of *access to high quality public green spaces* and were split again according to the variable *SGRolesHousewifeFulfilling*. Variable importance measures indicate that *DiscriminationAge* has the highest variable importance followed by *DiscriminationSocialPosition, SGRelationsIncome*, and three other variables indicating discrimination experiences. *SGRolesHousewifeFulfilling* had a rather low value, this split should therefore be interpreted with caution.

#### 3.3.2. CIT Results

Running a CIT analysis with the 40 sex/gender covariates for the exposure variable *access to high quality public green space* resulted in a tree with four splits and five subgroups (see Figure 1). Initially the population was split by the participant’s self-rated financial situation (*SGRelationsIncome*). Those with a bad or moderate rating were sent to one branch of the tree to be split again by degree of urbanization (*SGRelationsUrbanisation*). These two subgroups showed the lowest prevalence of access to high quality public green spaces. Participants with a good or very good self-rated financial situation were sent down a second tree branch where they were split by their experiences with age discrimination (*DiscriminationAge*) and once again by their self-rated financial situation (*SGRelationsIncome*). Thus, three different subgroups were formed: participants who stated they had already been discriminated against because of their age, those who had never experienced age discrimination and rated their self-rated financial situation as good and those who rated their financial situation as very good and had never experienced age discrimination. For the latter group, the highest prevalence of *access to high quality green spaces* was reported. As can be seen in Figure 1, this is the only subgroup where the proportions shifted, so that more participants had *access to high quality green* than to *only lower quality green*.

Conducting a test for variable importance with *cforest* also identified *DiscriminationAge* and *SGRelationsIncome* as the most important variables, followed by *DiscriminationSocialPosition* (Figure 2). Four other variables indicating discrimination experiences were also above the threshold suggested by Strobl [46]. However, the variable *SGRelationsUrbanisation*, that produced a split in our reported tree, had a variable importance score slightly below the threshold, thus this split should be interpreted with caution. In general, conducting the *cforest* analysis several times with different seeds showed that the only variables that were consistently above the threshold were *DiscriminationAge, SGRelationsIncome* and *DiscriminationSocialPosition*, while all other variables were at times above and at times below the threshold.

### 3.4. Greenness in the Residential Environment (Subjectively Measured)

#### 3.4.1. CART Results

Running a CART analysis for the exposure variable *greenness in the residential environment* with the 40 sex/gender covariates did not yield any splits after pruning with the *minimum cross-validation error rule* (see Appendix A, pp. 14–16).

#### 3.4.2. CIT Results

Running a CIT analysis with the 40 sex/gender covariates for the exposure variable *greenness in the residential environment* resulted in a tree with seven splits and eight subgroups (see Appendix A, pp. 16–18). The first splitting variable chosen was *CareActivitiesGardening*, dividing the population into participants answering “not applicable” versus all others. As the answer category “not applicable” was chosen by participants without a garden, this first split can be considered as a division of participants with versus without a garden. Those without a garden were split again by the participants’ self-rated financial situation (*SGRelationsIncome*). This led to two subgroups: participants with a bad or moderate self-rated financial situation and participants with a good or very good self-rated financial situation, the latter having a higher proportion of very green in the residential environment. Participants with a garden were sent down a second tree branch where they were split by degree of urbanization (*SGRelationsUrbanisation*). Participants who lived in a city were sent to one branch and those who lived in suburban or rural areas were sent to the other. Those from the city were split once again by their self-rated financial situation (*SGRelationsIncome*), with participants with a good or very good self-rated financial situation again showing a higher proportion of very green in the residential environment. Participants who lived in suburban or rural areas were further split by *DiscriminationSocialPosition, SexAtBirth* and once again *SGRelationsUrbanisation*. The *cforest* analysis with 500 trees showed that only the three covariates *SGRelationsUrbanisation, CareActivitiesGardening* and *SGRelationsIncome* should be further investigated according to the variable importance score threshold suggested by Strobl [46]. Therefore, the lower splits of the CIT tree described here must be interpreted with caution, as they included two covariates (*DiscriminationSocialPosition* and *SexAtBirth*), which showed a variable permutation importance value that was not higher than the random variation around zero.

### 3.5. Greenness within a 300 m Buffer around the Residential Address (Objectively Measured)

#### 3.5.1. CART Results

Results from a CART analysis for the continuous exposure variable *greenness within a 300 m buffer* indicated a tree with four splits (see Appendix A, pp. 19–22) after pruning with the *minimum cross-validation error rule* (pruning with the *1-SE rule* would have led to a tree with two splits). The first split was performed by the degree of urbanization (*SGRelationsUrbanisation*). Participants living in a city were sent to one side of the tree and were split again according to *SGRolesSingleParentEqual*, forming the two subgroups with the lowest mean greenness. The rest of the population was split again into participants living in suburban and those living in rural areas (*SGRelationsUrbanisation*). While participants living in rural areas formed the subgroup with the highest mean greenness, participants in suburban areas were further split according to whether or not they have their own garden (*CareActivitiesGardening*), with participants having a garden showing a higher mean greenness. Variable importance measures indicated that *SGRelationsUrbanisation* and *CareActivitiesGardening* were by far the most important splitting variables. On the other hand, *SGRolesSingleParentEqual* had a comparably low value and this split should therefore be interpreted with caution.

#### 3.5.2. CIT Results

Results from the CIT analysis for the continuous exposure variable *greenness within a 300 m buffer* indicated a tree with six splits leading to seven final subgroups (see Appendix A, pp. 23–24). The primary split was based on the degree of urbanization *(SGRelationsUrbanisation*) sending participants living in a city to one branch and the rest of the population to the other one. The participants living in a city were split again according to the variable *SGRolesSingleParentEqual* resulting in the two subgroups with the lowest mean of greenness. Participants from suburban or rural areas were split once again based on *SGRelationsUrbanisation*. Participants from rural areas showed the highest mean greenness and were split again according to their school education (*SGRelationsSchoolEducation*). Participants living in suburban areas were further split by *CareActivitiesGardening*, with those without their own garden showing the lowest mean of greenness. Participants with a garden were split again based on their experiences with discrimination because of their sexual orientation (*DiscriminationSexualOrientation*). Conducting a variable importance analysis indicated that *SGRelationsUrbanisation* and *CareActivitiesGardening* had by far the highest importance. Four other variables were also above the threshold suggested by Strobl [46], however *SGRelationsSchoolEducation, SGRolesSingleParentEqual* and *DiscriminationSexualOrientation* were not amongst them, so that those three splits in the tree described above should be interpreted with caution. When conducting the *cforest* analysis with different seeds, the results were very stable for *SGRelationsUrbanisation* and *CareActivitiesGardening,* but also two other variables, *SGRelationsFamilySituation* and *CareActivitiesChildren*, were consistently above the threshold and always the third and fourth most important variables.

### 3.6. Greenness within a 1000 m Buffer around the Residential Address (Objectively Measured)

#### 3.6.1. CART Results

Running a CART analysis with the 40 sex/gender covariates for the continuous exposure variable *greenness within a 1000 m buffer* resulted in a tree with three splits and four subgroups after pruning with the *minimum cross-validation error rule* (pruning with the *1-SE rule* would have led to a tree with two splits).

As can be seen in Figure 3, the primary split was based on the degree of urbanization (*SGRelationsUrbanisation*) sending participants living in a city to one branch resulting in the subgroup with the lowest mean of greenness. Those living in suburban or rural areas were sent to the other branch and were split once again by degree of urbanization (*SGRelationsUrbanisation*). Participants living in rural areas had the highest mean of greenness whereas those living in suburban areas were split again by *CareActivitiesGardening*, with all participants without a garden, being sent to the subgroup with the second lowest mean of greenness. All participants with a garden were sent down the other tree branch resulting in the subgroup with the second highest mean of greenness. The variable importance measures showed that *SGRelationsUrbanisation* was by far the most important covariate, while *CareActivitiesGardening* was the second most important variable.

#### 3.6.2. CIT Results

Running a CIT analysis with the 40 sex/gender covariates for the continuous exposure variable *greenness within a 1000 m buffer* resulted in a tree with seven splits and eight subgroups (see Appendix A, pp. 28–30). The first two splits were both based on the degree of urbanization (*SGRelationsUrbanisation*), sending participants living in a city, those living in rural areas and those living in suburban areas down three different tree branches. Those living in a city were split again by *SGRolesSameSexEqual*, resulting in the two subgroups with the overall lowest mean of greenness. Participants living in rural areas were split twice by school education (*SGRelationsSchoolEducation*), resulting in the three subgroups with the overall highest mean of greenness. Participants living in suburban areas were split again by *CareActivitiesGardening*, dividing participants based on whether they had a garden or not. The subgroup of participants with a garden were split once more by discrimination based on their sexual orientation (*DiscriminationSexualOrientation*). The subgroup of participants without a garden had a lower mean of greenness compared to the other two subgroups in this tree branch. The variable importance measures showed that *SGRelationsUrbanisation* and *CareActivitiesGardening* were by far the most important covariates from in total seven variables that were above the variable importance threshold suggested by Strobl [46]. The covariate *SGRelationsSchoolEducation* that led to a split here was also above the threshold, but so were four other variables with similarly high variable importance values that did not produce a split. The splits based on *SGRolesSameSexEqual* and *DiscriminationSexualOrientation* must be interpreted with caution, as they showed a variable importance value that was not higher than the random variation around zero. When repeating the *cforest* analysis with different seeds, the results were only stable for *SGRelationsUrbanisation* and *CareActivitiesGardening* and the two variables with the next highest values: *SGRelationsFamilySituation* and *CareActivitiesChildren*.

### 3.7. Analyses with 53 Covariates for a Subsample of n = 2624 Participants

For the subsample INGER KORA FIT with 2624 participants, we re-ran all decision tree analyses with a total of 53 covariates as we had 13 additional covariates available for these participants (see Appendix A). Detailed results of these analyses can be found in Appendix A. Overall, the results from this subsample confirmed the results from the analyses with the whole sample.

For the exposure measure *access to high quality public green spaces*, the first and most important splits were again based on the variables *DiscriminationAge* and *SGRelationsIncome*, with the variable importance measures suggesting that other variables indicating discrimination experiences were also of importance. Although two of the 13 additional variables led to splits in the trees for this exposure measure (i.e. *HealthBehaviorAlcohol* in the CART tree and *SGRelationsMobility* in the CIT tree), the variable importance measures showed that these variables were less important and the splits should be interpreted with caution.

For the exposure measure *greenness in the residential environment,* as well as the two NDVI measures, the results of the subsample confirmed the importance of *SGRelationsUrbanisation* and *CareActivitiesGardening*. Interestingly, the *cforest* analyses for both of the NDVI measures showed that besides the most dominant *SGRelationsUrbanisation* and *CareActivitiesGardening*, three other variables were consistently above the threshold of random variation: as in the whole sample these included *SGRelationsFamilySituation* and *CareActivitiesChildren*, but also the additional variable *SGRelationsHouseholdMembers*.

## 4. Discussion

Sex/gender is a multidimensional, non-binary, structural category, and can be comprehensively described within the multidimensional sex/gender concept, which we recently developed within the INGER project [9]. In order to adequately integrate sex/gender into quantitative research, statistical methods which can incorporate a high number of covariates, as well as their possible interactions, are required. We showed that decision trees fulfil these requirements and can be used to explore the relevance of multiple sex/gender dimensions for an environmental exposure. Using the exposure to green spaces in the residential environment as an exemplary field, we found that none of our covariates operationalizing the individual sex/gender self-concept (i.e., the dimensions sex assigned at birth, current sex/gender identity, internalized sex/gender roles and externalized sex/gender expressions) defined distinct subgroups with respect to the exposure to green spaces. However, we identified meaningful subgroups based on covariates contributing to explain structural sex/gender relations (i.e., discrimination experiences and intersectionality-related social categories). Thus, structural aspects related to sex/gender seem to play the most important role for differences in exposure to green spaces.

### 4.1. Methodological Considerations

#### 4.1.1. CART vs. CIT

We applied two different decision tree algorithms, CART and CIT. Both methods have certain advantages. It has been shown that CART yields a slightly lower prediction error than CIT and therefore has a higher predictive accuracy [14]. However, as CART considers all splitting points of all covariates simultaneously it has been argued that CART may have a bias to select variables with many possible splitting points or missings [39]. CIT was explicitly developed to overcome this bias by using a two-step approach. Firstly, the best splitting covariate is selected by testing for independence of the exposure measure. Secondly, the best splitting point is determined [39]. Venkatasubramaniam [15] pointed out that CIT is simpler to use than CART as it requires less parameter tuning and no pruning. They further argued that another advantage of CIT is that it relies on a concept of statistical significance based on valid *p*-values, providing the ability to make formal statistical statements of the results [15]. However, as Nembrini [36] pointed out, this last argument can also be seen as a disadvantage for research that is in general critical of the null hypothesis testing framework. As Gass [20] elaborates, the decision whether to choose CART or CIT may also depend on the research question and purpose of the study, stating that CART is the best option when the goal is classification or prediction, while CIT is better when the goal is to find the covariates showing the strongest association with the dependent variable. However, both are useful to identify complex interactions [20], which was our focus considering our intersectional perspective. Our results showed that if there is a covariate that clearly leads to the best split, as is the case for *SGRelationsUrbanisation* and *CareActivitiesGardening* in the analyses for the two NDVI exposure measures, or *SGRelationsIncome* and *DiscriminationAge* for the exposure access to high quality public green, then this split is mostly identified by both CART and CIT. However, in one case, i.e., the subjectively measured greenness in the residential environment measure, CART led to no splits after pruning, while CIT identified *SGRelationsUrbanisation, CareActivitiesGardening* and *SGRelationsIncome* for splitting. In this case CART was too conservative, as CIT identified meaningful splits. In general, our results showed that when using the *1-SE rule* for pruning CART led to no surviving splits, except for the first two splits based on *SGRelationsUrbanisation* for the two continuous NDVI exposure variables. This is rather conservative, and one might miss meaningful splits such as the ones based on the variable *CareActivitiesGardening*. Venkatasubramaniam [15] came to the same conclusion in their simulation study, recommending using the less conservative *minimum cross-validation error rule* for pruning. Overall, CIT led to trees with more splits and subgroups, however the *cforest* analyses with the variable importance measures then often showed that the additional splits must be interpreted with caution. The main conclusions from our results are therefore mostly the same for both CART and CIT.

#### 4.1.2. Considering Variable Importance Measures

Our results additionally showed that it is fundamental to consider variable importance measures when interpreting the results of single decision trees. As has been elaborated, single trees can be unstable and may be influenced by small changes in the sample [14,46]. Sometimes the choice between two similar variables depends on minor differences in samples and only considering the covariates included in the final tree can undermine the importance of additional covariates that have a meaningful influence on the dependent variable and would appear in trees in a slightly different sample. For example, in our analyses for the exposure variable *access to high quality public green space*, both CART and CIT included the covariate *DiscriminationAge*, which might imply that this variable is of particular importance. However, a closer look at the variable importance measures revealed that other discrimination variables, especially *DiscriminationSocialPosition*, were of similar importance. On the other hand, we often identified a split in our single trees, only for the variable importance measures to show that the splitting variable is of low importance, suggesting that this split may not be reproducible in a slightly different sample. Researchers should therefore avoid reporting only a single tree as their main result without giving any further information about variable importance.

#### 4.1.3. Exploratory Research

In general, decision tree analyses are exploratory and not suited to test hypotheses [16]. In our analyses we used covariates based on a newly established sex/gender questionnaire. We therefore had to use exploratory methods, as we were investigating new associations and did not have any a priori hypotheses about which sex/gender dimension may be of relevance for the availability of or access to green spaces. Decision trees are ideal to potentially find subgroups with particularly high or low exposure levels and to identify interactions that may not have been considered before. Nevertheless, we should keep in mind that this is hypothesis generating research and any conclusions must be tested in an independent sample in a next step.

#### 4.1.4. Benefits of Decision Trees

We identified several benefits of using decision trees for future analyses including multiple sex/gender covariates. Firstly, decision trees can deal with many covariates simultaneously, which is decisive for our multidimensional sex/gender approach [9]. Secondly, decision trees can identify subgroups defined by different combinations of covariates, which is essential for our intersectional approach [9]. While it is possible to include interaction terms in regression models, they can become very difficult to interpret when more than two variables are assessed at the same time [19]. For instance, for a higher number of covariates, a priori knowledge is required since it becomes necessary to pre-specify what interactions to test for [14]. This may lead to missing important interactions not thought of beforehand [13]. In addition, while decision trees have the ability to segment populations into meaningful subgroups, standard regression models focus on the effect of a covariate on the average member of a population [19]. Moreover, decision tree methods are non-parametric and no distributional assumptions have to be checked. Another important advantage of decision tree methods compared to standard regression models is that they can handle missing values in the covariates without excluding participants. This becomes more important the higher the number of covariates is, as the number of participants with at least one missing value increases. A complete case analysis as in classical regression usually excludes too many participants. For instance, in our case sample size would be reduced to N = 2534, leaving a third of the observations unused. Finally, decision trees are easy to interpret and intuitive. The visualization allows the reader to easily capture the subgroup structure and directly compare the different outcome distributions.

### 4.2. Relevance of Sex/Gender Dimensions for the Exposure to Green Spaces

For the exposure measure access to public green spaces, no subgroups characterized by sex/gender were found, which could be expected due to the homogeneous distribution of this exposure measure with 90.4% of the study population having access to public green spaces. Previous research in the United States suggests that white and wealthier communities often had more access to urban public green spaces such as parks whereas racial or ethnic minorities and people with low-income had less [32]. Studies in cities of the Global South also found similar results [30]. In contrast to studies in large cities, KORA participants live in the medium sized city of Augsburg (approx. 300,000 inhabitants) or its surrounding rural areas. In general, our sample had a rather high exposure to public green spaces and nearly 80% of the participants had access to a garden.

Regarding access to high quality public green spaces, both CART and CIT identified the self-rated financial situation (*SGRelationsIncome*) and discrimination experience based on age (*DiscriminationAge*) as best splitting variables. A closer look at the variable importance measures revealed that other discrimination variables, especially *DiscriminationSocialPosition*, are of similar importance and also would have split the group into participants with discrimination experience versus participants without. Thus, it seems that discrimination experiences based on different reasons, as assessed by our questions, were of similar importance for the access to high quality public green spaces. Participants with the highest prevalence of access to public green spaces of high quality rated their financial situation as very good and had no discrimination experiences, as assessed by our questions. These results were in line with previous research suggesting that the quality of parks is lower in areas of low-income earners and ethnic minorities than in wealthier neighborhoods [51,52]. It is interesting that we were able to detect results with the same tendency in our sample, even though the social divide between our participants was not as pronounced as in other studies, e.g., only 1.8% of our participants rated their financial situation as bad. Nevertheless, the slight difference between rating one’s financial situation as very good compared to good already influenced access to high or lower quality parks.

Our final three exposure measurements reflected the general greenness in the residential environment, either subjectively measured or using NDVI data with two different buffers. For all three analyses, two covariates were identified as the most important: the degree of urbanization (*SGRelationsUrbanisation*) and the covariate *CareActivitiesGardening*. We chose our covariates based on our multidimensional sex/gender concept [9]. *CareActivitiesGardening* was initially intended to capture the amount of work participants invest in gardening, in a sense of caring about work that must be carried out around the house. However, as the question = also included the answer option “not applicable” for participants without a garden, we unintentionally gave the decision tree algorithm the option to split participants into a subgroup having a garden and a second group without a garden. Thus, the variable *CareActivitiesGardening* turned into a proxy for the ownership of a garden. *SGRelationsUrbanisation* was included as a contextual intersectionality-related social category with the intention of identifying possible interactions between the dimensions representing the individual sex/gender self-concept and the degree of urbanization. Not surprisingly, splits based on these two covariates revealed the following results: participants in cities had the lowest amount of greenness, with participants in the rural areas having the most. Participants with a garden had higher greenness scores than participants without. Although this may not lead to new hypotheses regarding the relevance of sex/gender dimensions for the exposure to greenness, it verified the ability of decision trees to find meaningful splits in the data. Moreover, one has to keep in mind that *CareActivitiesGardening* did not lead to any splits in the trees for the two exposure variables capturing the access to (high quality) public green spaces. This result again showed the meaningfulness of the trees, since having a garden did not directly influence the access to public green spaces. For the subjectively measured greenness variable, but not for the NDVI measures, other meaningful splits were based on the covariate *SGRelationsIncome*, showing higher subjective greenness ratings for participants who rated their financial situation as good or very good. Another difference between the subjective and objective measures was, that for the subjective greenness the first split was based on *CareActivitiesGardening*, while for the two NDVI measurements the first two splits were elicited by *SGRelationsUrbanisation*. Thus, having a garden or not was most decisive for the subjective feeling of greenness in the residential environment, while the degree of urbanization had the biggest influence on the objectively measured greenness in a 300 or 1000 m buffer.

For both of the NDVI measures the random forest analyses identified further informative covariates, i.e., *CareActivitiesChildren* and *SGRelationsFamilySituation*, as well as the additional variable *SGRelationsHouseholdMembers* in the KORA FIT sample. However, none of these variables led to splits in the single tree analyses, thus we could not directly interpret their influence on the objective exposure to green spaces. Importantly, in our KORA sample, the covariate *CareActivitiesChildren* mostly referred to care activity for grand-children as the sample consisted of mainly older adults (mean = 63.4 years, ranging from 43 to 92 years). Therefore, our sample hardly included participants with young children, i.e., participants in the reproductive phase of their life. The age structure of the sample might have had an impact on the relevance of some of the sex/gender covariates, as the reproductive phase, i.e., the period of gender-segregated, unpaid family care work, represents an important part of the gender inequality relations [53]. For future studies it would be interesting to examine whether covariates such as *CareActivitiesChildren* might have had a greater influence on the final results in samples including more participants in the reproductive phase.

Taken together, in our study primarily structural aspects related to sex/gender, such as material situation and urban/rural context, but no covariates representing the individual sex/gender self-concept were relevant for differences in exposure to public green space or greenness in the residential environment. This is in line with current evidence on social inequalities in environmental exposures such as green spaces [30,31,32].

### 4.3. Future Research

Previous research on possible sex/gender differences focused on the usage of public green spaces and parks, discussing the relevance of differences between women and men in care activities for children or the elderly [54] or in concerns about personal safety [33,35]. Future research exploring the relevance of multiple sex/gender dimensions for the usage of public green spaces may therefore add to these results obtained with a binary sex/gender category.

Within the INGER project the next step will be to explore whether multiple sex/gender covariates modify the effect of green spaces on health, using another type of decision tree algorithm, i.e., model-based recursive partitioning [55]. Exploring subgroups with differential exposure-outcome relationships will allow us to move from the descriptive intersectional approach applied in the current study to a more analytic intersectional approach as defined by Bauer and Scheim [56].

### 4.4. Strengths and Limitations

As it is always the case for observational studies, our study has some limitations. The study population is rather homogeneous in terms of age, ethnicity, social disparities and availability of private gardens. Quality of public green spaces was rated subjectively by respondents and we defined “high quality” rather strictly based on responses to questions on maintenance and quality assessment. Furthermore, based on feedback from participants, we can assume that some exposure misclassification might have occurred when participants in rural areas negated having access to public green spaces, as they did not interpret privately owned forests or meadows as green spaces publicly available to them. Moreover, we only dealt with static green space exposure measures based on the participants’ place of residence, not considering human mobility, i.e., spatiotemporal changes of participants’ locations during daily routines [28,57].

The strengths of our study are the well-characterized KORA study population and the high response rate of our KORA INGER survey, the comprehensive assessment of sex/gender dimensions with at least 40 covariates based on a theoretically substantiated concept, and the subjective and objective measurement of exposure to green spaces or greenness. We also used a comparative statistical analysis applying two different decision tree algorithms, CART and CIT, with consideration of variable importance measures to avoid overinterpretation of spurious findings.

## 5. Conclusions

Most importantly, our study showed that decision tree analyses are suitable methods to analyze complex data in order to explore the relevance of multiple sex/gender dimensions for environmental exposures. With respect to our analyses exploring the relevance of multiple sex/gender dimensions for the exposure to green spaces, we showed that the participants’ financial situation and discrimination experience was relevant for their access to high quality public green spaces, while the urban/rural context was most important for the general greenness in the residential environment. The covariates operationalizing the individual sex/gender self-concept did not lead to homogeneous subgroups with respect to the exposure to green spaces. It is important to consider that this study was performed with a rather homogeneous study population in a non-metropolitan context. Further studies in larger urban areas with less private gardens and more different forms of public green spaces, and with a study population more heterogeneous with respect to age and social disparities, would add to the evidence on possible sex/gender exposure differentials. Moreover, besides exposure metrics, usage of green spaces should also be taken into account.

## Figures and Tables

**Figure 1 ijerph-19-07476-f001:**
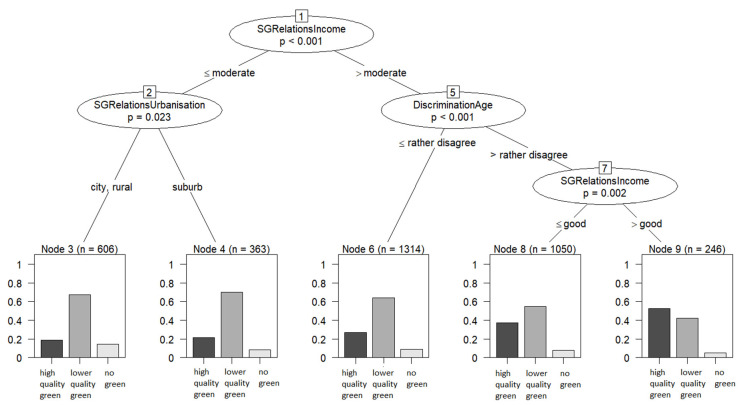
CIT tree for the exposure measure access to high quality public green space (subjectively measured). Bars in the bottom row show the proportion of participants in each exposure category.

**Figure 2 ijerph-19-07476-f002:**
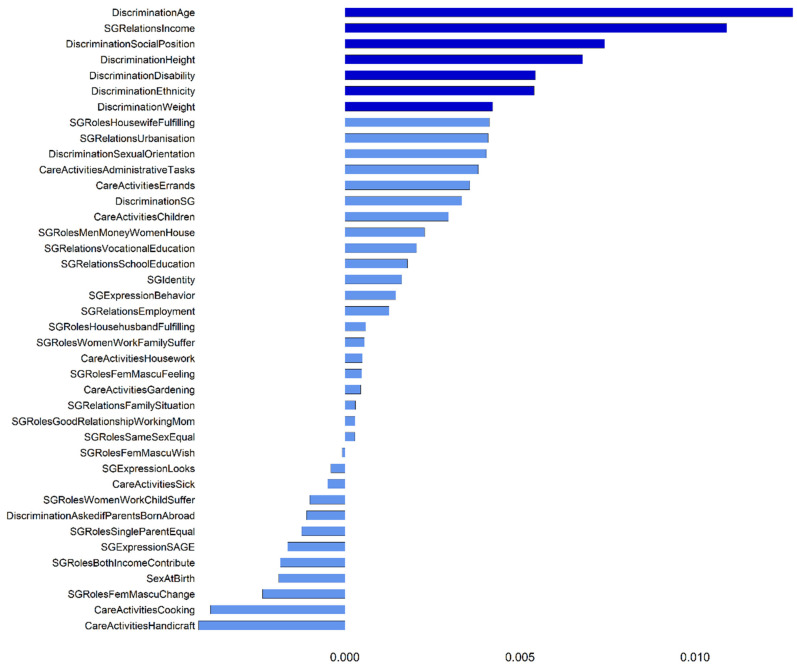
Variable importance measures calculated using random forest for the exposure measure access to high quality public green spaces. Values on the x-axis indicate mean decrease in accuracy for each covariate after random permutations. Bars colored in the darker shade correspond to variable importance values higher than the random variation around zero, a threshold for identifying informative variables suggested by Strobl [46].

**Figure 3 ijerph-19-07476-f003:**
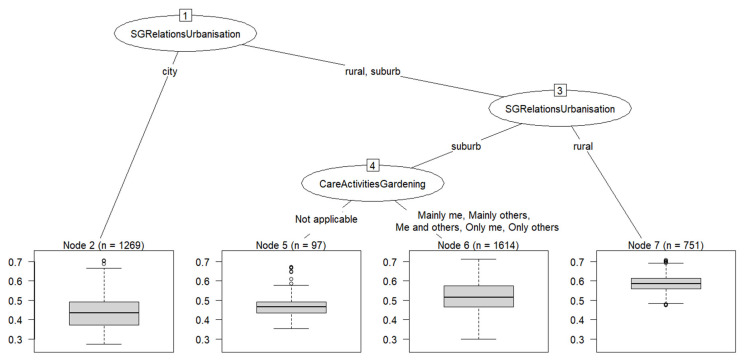
CART tree for the exposure measure greenness within a 1000 m buffer around the residential address (objectively measured). Boxplots in the bottom row show the distribution of the NDVI values in each subgroup.

**Table 1 ijerph-19-07476-t001:** The 40 Sex/Gender Covariates. Distributions within the INGER sample.

Covariate Name and Question * in INGER KORA Survey	Answer Categories and Distribution in the Whole Sample; *N* = 3742 (100%) ^#^	
**Individual sex/gender self-concept**		
** *Sex assigned at birth* **		
**SexAtBirth** What sex were you assigned at birth?	2014 (53.8) 1684 (45.0) 44 (1.2)	= female = male = missing
** *Current sex/gender identity* **		
**SGIdentity** What is your current sex/gender identity? (Multiple answers possible)	1998 (53.4) 1650 (44.1) 0 (0.0) 2 (0.1) 4 (0.1) 14 (0.4) 3 (0.1) 4 (0.1) 7 (0.2) 60 (1.6)	= female = male = intersexual = trans, trans man, trans woman = an identity not mentioned here = I do not want to classify as any SG category = female AND I do not want to classify as any SG category = male AND an identity not mentioned here = male AND I do not want to classify as any SG category = missing
** *Internalized sex/gender roles* **		
**SGRolesFemMascuFeeling** I mostly perceive myself as …	403 (10.8) 866 (23.1) 303 (8.1) 426 (11.4) 173 (4.6) 958 (25.6) 436 (11.7) 177 (4.7)	= very masculine = mainly masculine = a little masculine = just as feminine as masculine = a little feminine = mainly feminine = very feminine = missing
**SGRolesFemMascuWish** Ideally, I would like to be...	482 (12.9) 844 (22.6) 196 (5.2) 384 (10.3) 151 (4.0) 878 (23.5) 550 (14.7) 257 (6.9)	= very masculine = mainly masculine = a little masculine = just as feminine as masculine = a little feminine = mainly feminine = very feminine = missing
**SGRolesFemMascuChange** Has your assessment of what is feminine or masculine changed in recent years?	382 (10.2) 3206 (85.7) 154 (4.1)	= yes = no = missing
**SGRolesBothIncomeContribute** Both the men and women should contribute to the household income.	1766 (47.2) 990 (26.5) 801 (21.4) 95 (2.5) 33 (0.9) 57 (1.5)	= strongly agree = rather agree = neither agree nor disagree = rather disagree = strongly disagree = missing
**SGRolesMenMoneyWomenHouse** The man’s job is to earn money; a woman’s job is to look after the home and family.	106 (2.8) 293 (7.8) 898 (24.0) 1010 (27.0) 1380 (36.9) 55 (1.5)	= strongly agree = rather agree = neither agree nor disagree = rather disagree = strongly disagree = missing
**SGRolesGoodRelationshipWorkingMom** A working mother can establish just as warm and secure a relationship with her children as a mother who does not work.	1840 (49.2) 972 (26.0) 610 (16.3) 213 (5.7) 52 (1.4) 55 (1.5)	= strongly agree = rather agree = neither agree nor disagree = rather disagree = strongly disagree = missing
**SGRolesWomenWorkChildSuffer** A pre-school child is likely to suffer if his or her mother works.	424 (11.3) 1002 (26.8) 1112 (29.7) 730 (19.5) 414 (11.1) 60 (1.6)	= strongly agree = rather agree = neither agree nor disagree = rather disagree = strongly disagree = missing
**SGRolesWomenWorkFamilySuffer** All in all, family life suffers when the woman is working.	150 (4.0) 568 (15.2) 1377 (36.8) 993 (26.5) 596 (15.9) 58 (1.6)	= strongly agree = rather agree = neither agree nor disagree = rather disagree = strongly disagree = missing
**SGRolesHousewifeFulfilling** Being a housewife is just as fulfilling as working for pay.	520 (13.9) 622 (16.6) 1089 (29.1) 1013 (27.1) 430 (11.5) 68 (1.8)	= strongly agree = rather agree = neither agree nor disagree = rather disagree = strongly disagree = missing
**SGRolesHousehusbandFulfilling** Being a househusband is just as fulfilling as working for pay.	318 (8.5) 468 (12.5) 1045 (27.9) 1208 (32.3) 579 (15.5) 124 (3.3)	= strongly agree = rather agree = neither agree nor disagree = rather disagree = strongly disagree = missing
**SGRolesSingleParentEqual** One parent can raise a child as well as two parents together.	377 (10.1) 633 (16.9) 1181 (31.6) 1139 (30.4) 363 (9.7) 49 (1.3)	= strongly agree = rather agree = neither agree nor disagree = rather disagree = strongly disagree = missing
**SGRolesSameSexEqual** A same-sex couple can raise a child as well as a male-female couple.	667 (17.8) 838 (22.4) 898 (24.0) 708 (18.9) 567 (15.2) 64 (1.7)	= strongly agree = rather agree = neither agree nor disagree = rather disagree = strongly disagree = missing
** *Externalized sex/gender expressions* **		
**SGExpressionLooks** How would other people generally describe you based on your appearance, clothing style and other visual characteristics?	413 (11.0) 884 (23.6) 256 (6.8) 326 (8.7) 253 (6.8) 1012 (27.0) 388 (10.4) 210 (5.6)	= very masculine = mainly masculine = a little masculine = just as feminine as masculine = a little feminine = mainly feminine = very feminine = missing
**SGExpressionBehavior** How would other people generally describe you based on your behaviors?	384 (10.3) 876 (23.4) 301 (8.0) 455 (12.2) 232 (6.2) 953 (25.5) 338 (9.0) 203 (5.4)	= very masculine = mainly masculine = a little masculine = just as feminine as masculine = a little feminine = mainly feminine = very feminine = missing
**SGExpressionSAGE** Combination of SexAtBirth, SGExpressionLooks and SGExpressionBehavior SAGE-Score values: 1.0–7.0 low values = high socially assigned gender conformity high values = low socially assigned gender conformity	876 (23.4) 1767 (47.2) 508 (13.6) 303 (8.1) 43 (1.2) 5 (0.1) 4 (0.1) 236 (6.3)	= 1.0–1.5 = 2.0–2.5 = 3.0–3.5 = 4.0–4.5 = 5.0–5.5 = 6.0–6.5 = 7.0 = missing
**Items contributing to explain structural sex/gender relations**		
** *Experience of discrimination* **		
**DiscriminationSocialPosition** I have the feeling to be disadvantaged because of my position in society.	40 (1.1) 202 (5.4) 354 (9.5) 1379 (36.9) 1717 (45.9) 50 (1.3)	= strongly agree = rather agree = neither agree nor disagree = rather disagree = strongly disagree = missing
**DiscriminationAge** I have the feeling to be disadvantaged because of my age.	49 (1.3) 251 (6.7) 431 (11.5) 1343 (35.9) 1630 (43.6) 38 (1.0)	= strongly agree = rather agree = neither agree nor disagree = rather disagree = strongly disagree = missing
**DiscriminationHeight** I have the feeling to be disadvantaged because of my height.	23 (0.6) 71 (1.9) 143 (3.8) 922 (24.6) 2544 (68.0) 39 (1.0)	= strongly agree = rather agree = neither agree nor disagree = rather disagree = strongly disagree = missing
**DiscriminationWeight** I have the feeling to be disadvantaged because of my weight.	29 (0.8) 75 (2.0) 187 (5.0) 888 (23.7) 2521 (67.4) 42 (1.1)	= strongly agree = rather agree = neither agree nor disagree = rather disagree = strongly disagree = missing
**DiscriminationDisability** I have the feeling to be disadvantaged because of my physical impairment.	46 (1.2) 95 (2.5) 228 (6.1) 644 (17.2) 2681 (71.7) 48 (1.3)	= strongly agree = rather agree = neither agree nor disagree = rather disagree = strongly disagree = missing
**DiscriminationEthnicity** I have the feeling to be disadvantaged because of my ethnic/cultural affiliation.	14 (0.4) 25 (0.7) 56 (1.5) 408 (10.9) 3185 (85.1) 54 (1.4)	= strongly agree = rather agree = neither agree nor disagree = rather disagree = strongly disagree = missing
**DiscriminationSG** I have the feeling to be disadvantaged because of my sex/gender.	17 (0.5) 42 (1.1) 128 (3.4) 495 (13.2) 3018 (80.7) 42 (1.1)	= strongly agree = rather agree = neither agree nor disagree = rather disagree = strongly disagree = missing
**DiscriminationSexualOrientation** I have the feeling to be disadvantaged because of my sexual orientation.	13 (0.4) 6 (0.2) 25 (0.7) 251 (6.7) 3368 (90.0) 79 (2.1)	= strongly agree = rather agree = neither agree nor disagree = rather disagree = strongly disagree = missing
**DiscriminationAskedifParentsBornAbroad** Have you ever been asked in Germany whether you or your parents were born abroad?	415 (11.1) 3280 (87.7) 47 (1.3)	= yes = no = missing
** *Care activities* **		
**CareActivitiesChildren** Who is currently taking primary responsibility for the following tasks? (“Other people” also includes your partner) Care and/or upbringing of your children/grandchildren, driving services for your children/grandchildren	155 (4.1) 329 (8.8) 1173 (31.4) 393 (10.5) 71 (1.9) 1572 (42.0) 49 (1.3)	= only me = mainly me = me and other people = mainly other people = only other people = not applicable = missing
**CareActivitiesSick** Who is currently taking primary responsibility for the following tasks? (“Other people” also includes your partner) Care for disabled, chronically ill or in need of care family members, neighbors or friends	166 (4.4) 275 (7.4) 571 (15.3) 247 (6.6) 77 (2.1) 2342 (62.6) 64 (1.7)	= only me = mainly me = me and other people = mainly other people = only other people = not applicable = missing
**CareActivitiesCooking** Who is currently taking primary responsibility for the following tasks? (“Other people” also includes your partner) Cooking	1049 (28.0) 756 (20.2) 877 (23.4) 650 (17.4) 348 (9.3) 62 (1.7)	= only me = mainly me = me and other people = mainly other people = only other people = missing
**CareActivitiesHousework** Who is currently taking primary responsibility for the following tasks? (“Other people” also includes your partner) Housework	896 (23.9) 877 (23.4) 1200 (32.1) 629 (16.8) 98 (2.6) 42 (1.1)	= only me = mainly me = me and other people = mainly other people = only other people = missing
**CareActivitiesGardening** Who is currently taking primary responsibility for the following tasks? (“Other people” also includes your partner) Gardening (during the gardening season)	495 (13.2) 686 (18.3) 1449 (38.7) 423 (11.3) 111 (3.0) 535 (14.3) 43 (1.2)	= only me = mainly me = me and other people = mainly other people = only other people = not applicable = missing
**CareActivitiesErrands** Who is currently taking primary responsibility for the following tasks? (“Other people” also includes your partner) Errands (shopping, procurement)	797 (21.3) 800 (21.4) 1566 (41.9) 434 (11.6) 76 (2.0) 69 (1.8)	= only me = mainly me = me and other people = mainly other people = only other people = missing
**CareActivitiesAdministrativeTasks** Who is currently taking primary responsibility for the following tasks? (“Other people” also includes your partner) Administrative tasks (insurance, tax return, etc.)	1001 (26.8) 857 (22.9) 1191 (31.8) 481 (12.9) 163 (4.4) 49 (1.3)	= only me = mainly me = me and other people = mainly other people = only other people = missing
**CareActivitiesHandicraft** Who is currently taking primary responsibility for the following tasks? (“Other people” also includes your partner) Handicraft tasks in the household	704 (18.8) 936 (25.0) 906 (24.2) 850 (22.7) 303 (8.1) 43 (1.2)	= only me = mainly me = me and other people = mainly other people = only other people = missing
** *Intersectionality-related social categories* **		
**SGRelationsSchoolEducation** School education (Variable from basic KORA studies S1–S4)	1736 (46.4) 1085 (29.0) 920 (24.6) 1 (0.03)	= Degree after German basic secondary school (Hauptschulabschluss) = German O-levels (Mittlere Reife) = German A-Levels (Abitur) = missing
**SGRelationsVocationalEducation** Highest vocational qualification (Variable from basic KORA studies S1-S4)	272 (7.3) 2064 (55.2) 725 (19.4) 32 (0.9) 648 (17.3) 1 (0.03)	= no vocational qualification = vocational school/apprenticeship = technical school/master school = engineering school/polytechnical school = university of applied sciences/university = missing
**SGRelationsEmployment** Are you employed? If so: How many hours do you work on average per week (actual working hours)?	1970 (52.7) 22 (0.6) 132 (3.5) 218 (5.8) 249 (6.7) 642 (17.2) 323 (8.6) 95 (2.5) 91 (2.4)	= no = 1–5 h/week = 6–10 h/week = 11–20 h/week = 21–30 h/week = 31–40 h/week = 41–50 h/week = >50 h/week = missing
**SGRelationsIncome** How do you assess your financial situation?	394 (10.5) 2303 (61.5) 937 (25.0) 66 (1.8) 42 (1.1)	= very good = good = moderate = bad = missing
**SGRelationsFamilySituation** Do you live with a spouse or partner in a shared household? (Variable from KORA surveys F3, F4, FF4, FIT; the most recent available information from each participant was used)	2877 (76.9) 757 (20.2) 108 (2.9)	= yes = no = missing
**SGRelationsUrbanisation** Distribution of participants by degree of urbanisation.	1139 (30.4) 1392 (37.2) 751 (20.1) 460 (12.3)	= city = suburb = rural = missing

* Original questions were asked in German. ^#^ Percentages are calculated with respect to the whole sample, as participants with missing values in covariates are not excluded in the analysis.

**Table 2 ijerph-19-07476-t002:** Exposure distribution within the INGER sample.

Exposure and, if Applicable, Questions * in INGER KORA Survey	Answer Categories and Distribution in the Whole INGER Study Sample; *N* = 3742 (100%)	
**Subjective exposure measurement**		
**Access to public green spaces** Are there publicly accessible green spaces (e.g., parks, forests, meadows) in your neighbourhood?	3383 (90.4)	= yes
	334 (8.9)	= no
	25 (0.7)	= missing
**Access to high quality public green spaces** The green spaces in my neighbourhood are well maintained. The green spaces in my neighbourhood are of high quality. (Combination of three variables)	1066 (28.5)	= high quality green
	2179 (58.2) 334 (8.9) 163 (4.4)	= only lower quality green = no green = missing
**Greenness in the neighbourhood** How green is your neighbourhood? (From green strips along the road to gardens and parks.)	2911 (77.8)	= very green
	731 (19.5)	= little green
	78 (2.1)	= hardly green
	22 (0.6)	= missing
**Objective exposure measurement**		
**Greenness within a 300 m buffer around the residential address** Calculated from several satellite images between April and October in 2019. Negative pixels of the NDVI map were excluded prior to assignment to home addresses.	0.16 0.41 0.47 0.47 0.09 0.53 0.73 *N* = 11	= min = Q1 = median = mean = SD = Q3 = max missing
**Greenness within a 1000 m buffer around the residential address** Calculated from several satellite images between April and October in 2019. Negative pixels of the NDVI map were excluded prior to assignment to home addresses.	0.27	= min
	0.44 0.50 0.50 0.09 0.57 0.71 *N* = 11	= Q1 = median = mean = SD = Q3 = max missing

* Original questions were asked in German.

**Table 3 ijerph-19-07476-t003:** Further description of the INGER study population.

Question * In INGER KORA Survey	Answer Categories and Distribution in the Whole Sample; *N* = 3742 (100%)	
Age distribution in the INGER study population.	Continuous variable	(years)
	63.41	= mean
	9.42 43.00 92.00 0	= SD = min = max = missing
What is your current employment status?	1681 (44.9) 1786 (47.7) 184 (4.9) 91 (2.4)	= employed = retired = other = missing
Do you live…?	2951 (78.9) 759 (20.3) 32 (0.9)	= in your own property = for rent = missing
How long have you lived at your current address?	Continuous variable 29 years 16.31 0 93 68	(years) = mean = SD = min = max = missing
How often do you usually reside at your current address?	3655 (97.7) 19 (0.5) 25 (0.7) 17 (0.5) 26 (0.7)	= daily = only on weekdays = only on days off = few days a month = missing
Does your flat or house have a garden?	2614 (69.9) 362 (9.7) 723 (19.3) 43 (1.2)	= yes, for sole use = yes, shared with several parties = no = missing
Do you have a balcony and/or terrace?	2684 (71.7) 985 (26.3) 73 (2.0)	= yes = no = missing
Do you use your garden, balcony or terrace for recreation?	3359 (89.8) 287 (7.7) 63 (1.7) 33 (0.9)	= yes = no = neither garden, balcony nor terrace available = missing
During the summer months, how often do you visit publicly accessible green spaces, such as… … parks, forests, meadows, which you can reach in about 10 min?	289 (7.7) 283 (7.6) 264 (7.1) 528 (14.1) 1146 (30.6) 794 (21.2) 334 (8.9) 104 (2.8)	= (almost) never = 3–6 times per year = 7–10 times per year = at least once a month = at least once a week = (almost) daily = no publicly accessible green spaces available = missing
During the summer months, how often do you visit publicly accessible green spaces, such as… … parks, forests, meadows, which you cannot reach in about 10 min?	569 (15.2) 539 (14.4) 473 (12.6) 792 (21.2) 692 (18.5) 144 (3.9) 334 (8.9) 199 (5.3)	= (almost) never = 3–6 times per year = 7–10 times per year = at least once a month = at least once a week = (almost) daily = no publicly accessible green spaces available = missing

* Original questions were asked in German.

## Data Availability

The data that support the findings of this study are available from the KORA study team of the Helmholtz Zentrum München but restrictions apply to the availability of these data, which were used under license for the current study, and so are not publicly available.

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
