# Peer review of "Decision Tree Analyses to Explore the Relevance of Multiple Sex/Gender Dimensions for the Exposure to Green Spaces: Results from the KORA INGER Study"

_ijerph, 2022, doi:10.3390/ijerph19127476_

Round 1
Reviewer 1 Report
Thank you for giving me this opportunity to read the manuscript entitled "Decision tree analyses to explore the relevance of multiple sex/gender dimensions for the exposure to green spaces: results from the KORA INGER study". The topic of this manuscript is interesting and would be a good contribution to this field. I think it could be considered for publication in International Journal of Environmental Research and Public Health once the following issues are addressed.
- Please replace the keywords that already appear in the manuscript's title with close synonyms or other keywords, which will also facilitate your paper to be searched by potential readers.
- The GEE-based NDVI calculation is reasonable. I have a suggestion here (the authors do not need to revise the manuscript). Actually, there are algorithms in GEE that can be used to mask the cloud-cover area, which means you can take advantage of year-round images (or all the images collected between April to October) with cloud-cover masked. I hope this suggestion will be helpful for the authors' future research.
- The resolution of the images in the manuscript's figures is a little bit low.
- Human mobility is not considered in the greenspace exposure assessment, which will inevitably create uncertainty. This issues should be discussed in the Limitation section. The following papers are suggested to be cited as references: (1) "Dynamic assessments of population exposure to urban greenspace using multi-source big data", and (2) "Dynamic assessment of PM2. 5 exposure and health risk using remote sensing and geo-spatial big data".
- Some grammatical errors exist in the manuscript. Therefore, a critical review of the manuscript language will improve readability.
Reviewer 2 Report
Great research project. I have just two comments:
· What is INGER?
· It is interesting how the survey asked about discrimination based on culture or ethnicity but did not ask their culture or ethnicity. If the study population were mostly homogeneous group, this question has little impact on the results. However, if the population is a diverse group, then it would affect the results. I would suggest to include ethnicity data in the results and discuss them as well.
Round 2
Reviewer 1 Report
Thank you for giving me this opportunity to read the revised version of the manuscript titled "Decision tree analyses to explore the relevance of multiple sex/gender dimensions for the exposure to green spaces: results from the KORA INGER study", and for the detailed responses to my earlier comments. I am satisfied with this revised version, and I think it is acceptable now.